# What is the scope of teaching and training of undergraduate students and trainees in point of care testing in United Kingdom universities and hospital laboratories?

**Lee Peters**◉*, **Ana Sergio Da Silva**◉, **Philip Mark Newton**

Research in Health Professions Education, Swansea University Medical School, Swansea University, Swansea, Wales, United Kingdom

* 141750@swansea.ac.uk

**Data Availability Statement:** All relevant data are within the paper.

**Funding:** The author(s) received no specific funding for this work.

## Abstract

Point of care testing (POCT) is an analytical test performed by a healthcare professional outside of a conventional laboratory. The global POCT market was valued at US$ 23.16 billion in 2016 and is forecasted to grow to US$ 36.96 billion in 2021. This upward trend for POCT has increased workload for pathology departments who manage POCT. This research aims to characterize and analyse the teaching and training of POCT at United Kingdom (UK) universities on Institute of Biomedical Science (IBMS) accredited biomedical science degrees, and at UK hospital laboratories. A freedom of information (FOI) request was sent in 2018 to all 52 UK universities with an accredited IBMS Biomedical science degree to request information on teaching of POCT, with a 100% response rate. Further FOI requests were sent to all National Health Service (NHS) hospital pathology departments in the UK, regarding POCT training provided to trainee Biomedical scientists, with a 97% response rate. Twelve of the degrees contained no POCT teaching, with a further 9 having no specific POCT teaching. Sixty-six laboratories confirmed that there was no POCT training. The university teaching hours varied between 0 and 35 hours. The median time spent teaching POCT at university was 2 hours. The laboratory teaching hours varied between 0 and 450 hours The median time spent teaching POCT in hospital laboratories was 3 hours. A content analysis of the learning outcomes provided by 29 universities showed that only 61% (84/137) were measurable and 26% (36/137) of the learning outcomes used action verbs that have previously been listed to be avoided in learning outcome writing. Only 9% (13/137) of outcomes specifically described POCT, with 8 of these being measurable. The findings demonstrate that although this is a commonly required skill for biomedical scientists, there is a clear lack of POCT teaching and training in the UK. To meet the new Quality Assurance Agency for Higher Education (QAA) guidelines, but most importantly to ensure the workforce is fit for the needs of the current healthcare system, the quality and quantity of POCT teaching and training needs to improve.

**Competing interests:** The authors have declared that no competing interests exist.

# 1 Introduction

The United Kingdom (UK) Medicines and healthcare products regulatory agency (MHRA) define point of care testing (POCT) as any analytical test performed for a patient by a healthcare professional outside of the conventional laboratory setting [1]. Other terms commonly used to describe POCT include near patient testing (NPT) and bedside testing.

POCT use is increasing, with the global POCT market expected to grow from US$ 23.16 billion in 2016 to US$ 36.96 billion in 2021 [2]. This increasing demand has been attributed to increasing clinical demand, heavy industry promotion, short turnaround time for results, economical and practical factors, and advancements in technology [3]. Examples of POCT include glucose meters, pregnancy tests, International Normalised Ratio (INR) and Coronavirus disease of 2019 (COVID-19), severe acute respiratory syndrome coronavirus 2 (SARS-CoV-2). The global pandemic has exacerbated demands for POCT. To meet this growing demand, laboratory staff will need to be trained to support and deliver this service.

The growth and development of POCT has been identified in key UK government healthcare strategies. An example is 'Looking forward: Healthcare science in NHS Wales' [4]. This document published by the Welsh government gives POCT in the community as an example of a new role for healthcare scientists. In 2019 the Welsh government also published the pathology statement of intent [5]. This document lists POCT as a pathology specialty and that it is one of the fields that is "... *rapidly evolving, driven by a reduction in costs, increased connectivity and technological innovation*". The statement of intent also discusses how POCT will be part of the service design of pathology in Wales. The Scottish healthcare science national delivery plan 2015–2020 [6], lists POCT as one of the five service improvement programmes that will "... *deliver high-quality, sustainable health and care services for Scotland...*". The NHS England long-term plan [7] discusses the use of POCT as a method to prevent unnecessary hospital admissions. The 'Science in Healthcare: delivering the long-term plan report' [8] from the chief scientific officer of England, lists POCT as an emerging technology that will shape healthcare science and a key component in care pathways.

## 1.1 Biomedical science education in the United Kingdom

The health and care professions Council (HCPC) is the regulator of 15 health and care professions in the UK. The professions include physiotherapists and radiographers. Biomedical scientist (BMS) is the protected title of a laboratory scientist that has met the HCPC standards of proficiency. Fig 1 shows the routes to HCPC registration for BMS.

The institute of biomedical science (IBMS) certificate of competence (COC) is a qualification that demonstrates the student/trainee meets the HCPC standards of proficiency for registration as a BMS [10]. In Fig 1 routes one and two, students will undertake an IBMS accredited undergraduate degree and laboratory training either during or after completion of the degree. Route 3 is for students that do not have an accredited degree and require educational 'Top-up's', to ensure they meet the educational requirements, and clinical laboratory training. Route 4 is an equivalence route to registration. This route is not part of this research due to low numbers and a non -standardised training program.

## 1.2 Institute of biomedical science degree accreditation and the United Kingdom Quality Assurance Agency

Undergraduate degree accreditation by the IBMS ensures the degree covers the specified subjects, in specific depth, to meet the HCPC standards of proficiency for BMS. Degrees are only accredited if they incorporate the requirements detailed in the Quality Assurance Agency for

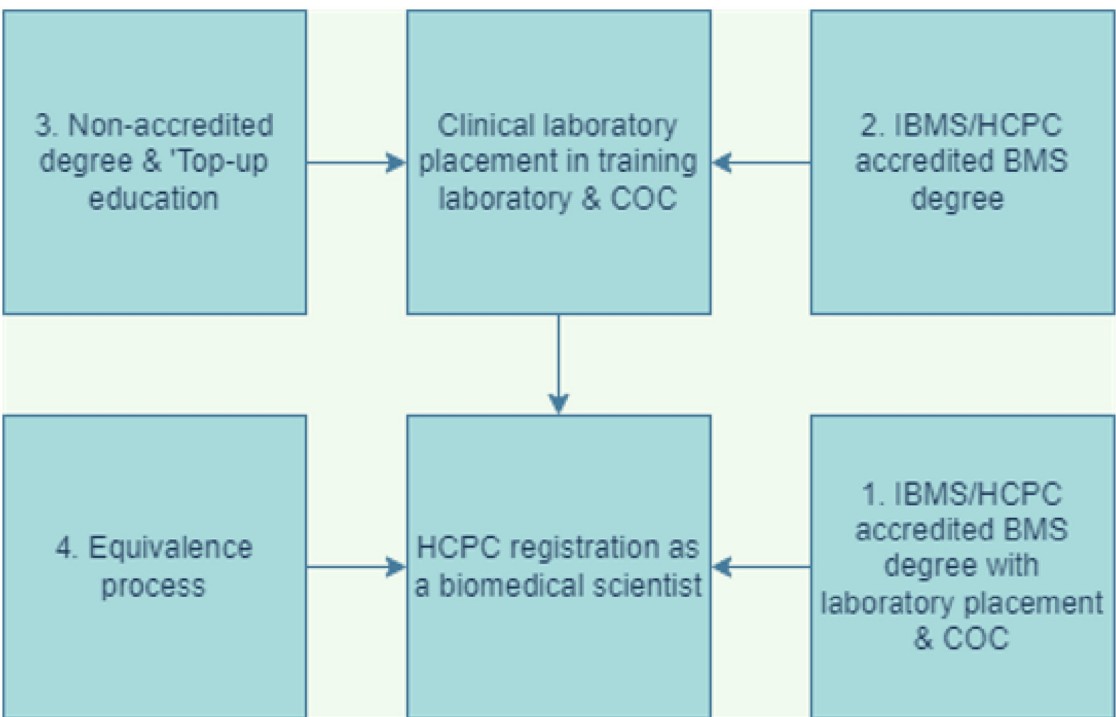

**Fig 1. The four routes to health and care professions council (HCPC) registration for biomedical scientists 'Top-up' education: Refers to supplementary education needed to meet the HCPC requirements.** Equivalence process: refers to the route used for laboratory scientists that work at the level of BMS and want to register as a BMS but that are not HCPC registered, (adapted from information from Institute of Biomedical Science on becoming HCPC registered [9]). COC: certificate of competence; IBMS: institute of biomedical science; HCPC: health and care professions council, BMS: biomedical scientist.

higher education (QAA) subject benchmarking for biomedical science [11]. The accreditation process also looks at other aspects of the program, such as, program delivery and placement opportunities [12].

The QAA is an independent body that monitors and advises on standards and quality in the UK higher education. QAA benchmark statements define the academic standards that can be expected of a graduate, in terms of what they might know, do and understand at the end of their studies and describe the nature of the subject [11]. The QAA initiates regular reviews of their content, or in response to significant changes in the discipline [11]. They do not provide a national curriculum for a subject but provide general guidance for writing the learning outcomes (LOs) of the course.

POCT is referenced twice in the HCPC standards of proficiency for BMS:

1. Standard 14.16, know the extent of the role and responsibility of the laboratory with respect to the quality management of hospital, primary care and community-based laboratory services for NPT and non-invasive techniques.

2. Standard 14.26, be able to use standard operating procedures for analyses including POCT in vitro diagnostic devices

Students or trainees undertaking the COC must meet all the standards of proficiency, including the standards shown in Table 1. To become IBMS accredited, undergraduate courses only have to meet the QAA benchmarking statement.

**Table 1.** The survey questions which were used to assess teaching and training of point of care testing at universities (survey 1) and national health service laboratories (survey 2).

| Survey 1 (n = 52) | Survey 2 (n = 179) For each pathology discipline in your trust/health Board (i.e. microbiology, haematology, blood transfusion, biochemistry, histopathology etc. . .); |
|---|---|
| 1. Please could you list all your Institute of biomedical science (IBMS) accredited courses your institution currently run? | 1. On average how many trainee biomedical scientist/ university biomedical science placement students do you have a year? |
| 2. Approximately how many hours of teaching time is dedicated to point of care testing/ Near patient testing on each of your institute of biomedical science (IBMS) accredited courses? | 2. What types of evidence are used by your students/ trainees to meet the Health and Care Professions Council (HCPC) standards of proficiency for biomedical scientists for standards 14.26 and 14.16 in the certificate of competence i.e. reflective sheet, essay, competency etc. . .? |
| 3. Please could you provide any learning outcomes relating to point of care testing/ near patient testing and how are they assessed? | 3. Approximately how many hours of teaching/training is dedicated to point of care testing / near patient testing (NPT) for each student/trainee? |
| 4. Please could you provide the learning outcomes used to meet the health and care professions council (HCPC) Standards of proficiency for biomedical scientists for standards 14.26 and 14.16 and how are they assessed? | 4. What types of teaching/training do you give these students/trainees in point of care testing i.e. seminars, practical training etc. . .? |
| | 5. Do your students/trainees get a secondment/rotation into a point of care section and if so, for how long? |

The QAA benchmark statement released late 2019 [11] now includes NPT under the clinical chemistry requirements. All universities seeking IBMS accreditation or re-accreditation post 2019 will have to evidence how they meet the new benchmark statement. NPT has been added to the benchmark statement *"the principles and applications of biochemical investigations used for screening, diagnosis, treatment and monitoring of disease, including near-patient testing"*. POCT has been added *"in light of recent advances in practice"*. [11]

POCT was not specifically mentioned in the previous QAA biomedical science statement [13]. This research was conducted before the new QAA benchmark statement was released in late 2019. Therefore, universities who responded to the FOI request were accredited under the 2015 benchmark statement.

## 1.3 Learning outcomes

LOs are a statement of what a learner is expected to know and be able to demonstrate by assessment after learning. Gosling and Moon [14] state that;

> *"Well written learning outcomes provide a means of mapping the content of a curriculum— for example, to see how they reflect benchmark statements, which of the key skills are acquired, where the same skill or content is appearing more than once in the programme, the capabilities the students acquire as they progress through the levels in the programme of study."*

There appears to be no standard definition of a LO, several different definitions have been suggested, see S1 Appendix [14–19]. Whilst there is no standard definition there is a conserved theme. Kennedy et al. [19], stated that the various definitions they cited did not differ significantly and it was clear that LOs focus on what the learner has achieved and what the learner can demonstrate at the end of a learning activity.

The use of LOs has been reported to have many benefits. They enable internal and external quality assurance to determine how appropriate the curriculum is [14]. The LOs allow students to gauge what they can achieve if they participate on a module/course. Mahajan and Sarjit Singh [20] listed a range of perceived benefits that in their opinion included; helping the student choose an appropriate course, give a clear idea to the students of what are they going to learn or achieve at the end of the class before the start of every class, help teachers design their teaching material more effectively, help teachers select appropriate strategies for teaching and make assessments mapping clear and easy. Given the importance of LOs throughout the years many theorists have suggested approaches to their design. Some suggested structures for a good LO are described in S2 Appendix [19, 21–24].

LOs use verbs to describe the expectations of the student. Bloom's Taxonomy, first published in 1956 [25] and revised in 2002 [26], is often used to write LOs. It classifies thinking skills into six hierarchically organized categories that range from lower-level (know and understand) to higher-order (apply, analyze, evaluate, create) [25]. Each of the levels have indicative verbs. Stanny [27], and in previous work [28], analyzed examples of verb lists on educational websites in the United States of America and the UK respectively. Both studies found little consistency in the lists, including several verbs that appeared in all levels of the hierarchy suggested by Bloom. We have previously suggested a list of verbs to use at each hierarchical level and a list of verbs to avoid using in writing LOs [28].

## 1.4 Freedom of Information (FOI) requests

The UK FOI act (2000) [29] and FOI (Scotland) act (2002) [30] gives the right to access recorded information held by public sector bodies. The FOI act has some limitations about what information is covered by the act, e.g. cost above the allowable threshold, repeat of a previous request or the request is vexatious. To make a FOI request, the organisation must be contacted in writing, e.g. e-mail, letter, online form and requires your name, contact details and detailed description of the information required. The legislation allows 20 working days for a response.

FOI requests have been used as a mechanism to distribute surveys, questionnaires etc. to gather data for research. Fowler et al. [31] conducted a systematic review of the use of the UK FOI act to gather data for healthcare research. The 16 studies reviewed covered a range of requests such as litigation, surgical provision and laboratory provision. One of the studies included in the review tried to obtain data via a letter and achieved a response rate of 11%. When they made the same request via FOI the response rate rose to 83%. FOI requests have also been used in educational research. Knight et al. [32] used a FOI to look at the academic performance of graduate-entry and undergraduate medical students at a UK medical school.

Phillips et al. [33] reported a response rate between 26.6% and 100%, with a mean of 71.3% (SD 19.5%), when they looked at survey response rates in three education journals. They compared this to other reported response rates; United States of America government surveys (~ 70%), and other surveys across Europe (50–60%). Health profession education research using FOI requests have shown a response rate of 97%, 93%, 83% (Arulrajah and Steele [34], Ayton and Ibrahim [35], Copeland and Barron [36]) respectively.

The research presented here is a pragmatic approach to answer the research questions to inform further research into the BMS curriculum and laboratory training with regards to POCT teaching and training.

The current state of POCT teaching in universities and clinical laboratories is unknown and this research seeks to answer the following research questions:

1. How much POCT is taught in UK accredited undergraduate biomedical science degrees pre change in QAA benchmark statement?

2. How POCT is taught in UK accredited undergraduate biomedical science degrees pre change in QAA benchmark statement?

3. How much POCT is taught in UK pathology laboratories pre change in QAA benchmark statement?

4. How POCT is taught in UK pathology laboratories pre change in QAA benchmark statement?

## 2. Materials and methods

A survey was sent to all UK universities offering IBMS accredited biomedical science degrees in May 2018, with second survey sent to all the NHS trusts/health board laboratories in the UK in October/November 2018. The surveys were sent using a FOI request. The wording for each survey is provided in Table 1.

The questions used in the first survey seek to gather information on time allocated to POCT teaching at the selected universities and the topics covered. The questions used in the second survey seek to gather information on how the pathology laboratories train their trainee BMS on POCT including how they met the relevant HCPC standards. The separate surveys were done in order to assess the different avenues of HCPC registration of trainees/students (refer to Fig 1). The participants in survey one complete an IBMS accredited degree (Fig 1, pathways 1 and 2) whilst those of survey 2 achieve a COC during a clinical placement or training position. The second survey asked the laboratories to provide information for all relevant disciplines in their hospital trust or health board. This was to capture practices per pathology discipline, if there was no pan pathology approach to training.

The surveys were sent a few months apart due to the large number of requests being sent out and to allow time to deal with queries.

### 2.1 Selection process

A list of UK-based IBMS-accredited degrees was obtained from the IBMS website [37]. Only UK universities were selected as the aim of the research was to identify POCT training in the UK. This resulted in 52 university FOI requests.

Despite an extensive search, there was no list of pathology laboratories in the UK. Pathology laboratories are the main provider of employment of BMS. Therefore, an FOI request was sent to every NHS acute trust or health board in the UK. To identify these, the websites for NHS England, Wales, Scotland and Northern Ireland were searched. Additionally, the public health and blood transfusion trusts for each nation were also added to the list. The trusts and health boards were selected as they host pathology as part of their organisation. This identified 179 institutions (trusts/healthboards) where FOI requests could be sent.

The FOI requests were sent to each university/trust/health board dependent on the instructions on their internet site.

### 2.2 Data analysis

All responses were recorded in excel for analysis. Each response was anonymised and given a unique identifier to avoid any potential reputational damage to the universities and laboratories. Response rate, response turnaround time, teaching and training hours (range, median and interquartile range) were calculated. If range of training hours was provided, then the maximum value was used.

LOs were reviewed and assessed against the following criteria:

1. Was the LO measurable?

2. Does the LO specifically describe POCT?

3. Does the LO have multiple outcomes?

4. Is the action verb used on the recommended/avoid list as described by Newton et al. [28]?

5. What level of blooms taxonomy would the verb be mapped to?

The LOs were assessed by at least one author and then reviewed by the other two authors. Any changes to the level of taxonomy assignment required agreement by all three authors.

To assign the verb to a level of blooms taxonomy, the authors used the supplementary data from Newton et al. [28]. The supplementary data includes the verb lists from 47 UK higher education institutions. Verbs were assigned to the level where it appeared in the majority of these lists.

For training hours, a day was considered to be 7.5 hours and a week 37.5 hours unless otherwise stated. When calculating median, if the respondent gave a range of time, then the highest value was used to calculate the mean. Unknown responses were not included in the calculation.

### 2.3 Ethics

Whilst the FOI acts allow anyone to submit FOI requests, ethics approval was obtained for the use of the FOI system in order to gather research data and the subsequent analysis and publication. (Swansea Medical School Research Ethics Committee (Project reference number 2018–0030)).

## 3 Results

### 3.1 Response rate & turnaround time

For the first survey to the universities offering IBMS accredited undergraduate degrees, 100% (52) responded, while for the second survey to the UK trusts/health boards with clinical laboratories, 97% (173/179) responded.

77% (40) of the universities responded within the 20-working day limit, whilst 84% (146) of the trusts/health boards who responded did so within the limit.

### 3.2 Point of care testing training in universities

Fifty-two universities responded and gave results for 55 accredited degree programs. Of these, nine (16%) described their POCT teaching as embedded within the curriculum and so the teaching hours could not be quantitated. Twelve programs (22%) stated that there were no specific teaching hours dedicated to POCT.

The amount of direct teaching, non-embedded, received by an undergraduate on POCT ranged from 0 to 35 hours with a median (IQR) of 2 hours (0.25–5.75). Fig 2 shows the range of POCT teaching hours provided by the universities in response to question 2 of survey 1.

When asked to provide the LO for teaching on POCT (survey 1 question 3), 29 universities (56%) responded with at least one LO. The range of LOs provided per university was 1–31 with 137 LOs provided in total. There were 192 action verbs, of which 54 were unique (see S3 Appendix).

The analysis found that of the 137 LOs, only 61% (84/137) were considered measurable. Thirteen of the LOs provided specifically mention POCT and of these 8 were measurable. See Table 2 for summary of analysis.

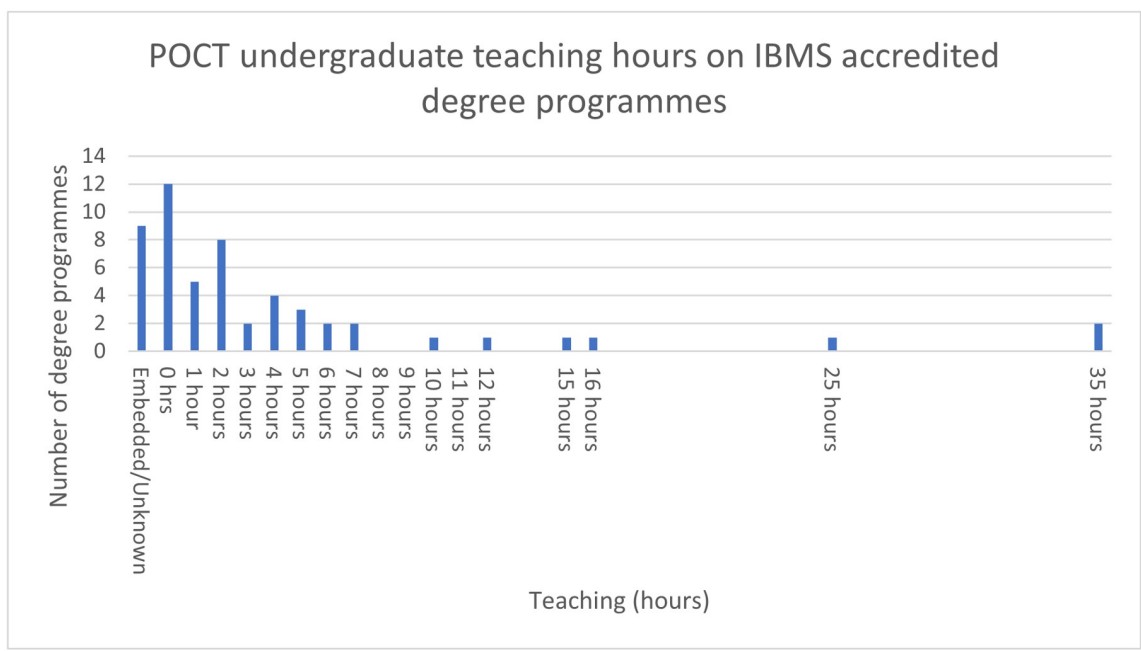

**Fig 2. Point of care testing teaching hours by number of degrees programmes on institute of biomedical science accredited degree in the United Kingdom 2018.**

Forty-seven of the provided LOs had multiple outcomes described. Forty-two of these had 2 outcomes, 1 had 3, and 4 had 4 outcomes described per LO.

We also considered whether the LO action verb appeared on the recommended/avoid list described by Newton et al. [28]. Verbs on the recommended list appeared in 49% (67/137) of LOs, whilst 26% (36/137) of the LOs had an action verb from the avoid list, see S4 Appendix.

The level of blooms taxonomy was assessed for each action verb. Of the 54 unique verbs used, 37 could be assigned to a level on blooms taxonomy, 7 were on the avoid list and 9 verbs were not found on any of the lists described by Newton et al. [28]. One verb, decide, was unmapped as it was split between the synthesis and evaluation levels, with 4 lists placing it on the synthesis and 4 on the evaluation level, see Table 3. For the full list, please see S4 Appendix.

### 3.3 Point of care testing training in national health service laboratories

Table 4 shows the evidence used by clinical laboratories to meet requirements for HCPC standards of proficiency 14.16 and 14.26, provided in response to question 2 of survey 2 (Table 1). Of the 173 laboratories surveyed, 118 responded. Of these, 76 gave a response for their whole organisation, with the 42 remaining laboratories giving discipline specific responses (i.e.

**Table 2. Analysis of the 137 point of care testing (POCT) learning outcomes from universities providing institute of biomedical science accredited degrees in the United Kingdom.**

|  | Is the LO measurable? | Does the LO describe POCT specifically? | Does the LO contain multiple outcomes? | Does the LO contain an action verb on the Recommended list? |
|---|---|---|---|---|
| Yes | 84 (61%) | 13 (9%) | 47 (34%) | 67 (49%) |
| No | 53 (39%) | 124 (91%) | 90 (66%) | 93 (68%) |

LO: learning outcomes. POCT: point of care testing.

**Table 3. Action verbs mapped to level of Bloom's taxonomy.** The verbs were assigned using the lists collated by Newton et al. [28].

| level of blooms taxonomy | Count of verbs assigned to level |
|---|---:|
| Evaluation | 3 |
| Unmapped | 1 |
| Synthesis | 8 |
| Analysis | 5 |
| Application | 9 |
| Comprehension | 6 |
| Knowledge | 6 |
| Not on list | 9 |
| Verbs to Avoid | 7 |

biochemistry, microbiology). This resulted in 181 usable responses. These laboratories used 15 different types of evidence to meet the specified standards. For a breakdown between the whole trust responses and the different pathology disciplines, see S5 Appendix.

For the total number of hours spent training on POCT (Table 1, survey 2, question 3), 147 out of the 173 laboratories responded. Of these, 81 gave a response for their whole organisation, with the 66 remaining laboratories giving discipline specific responses (i.e., biochemistry, microbiology). This resulted in 254 usable data points. Table 5 summarizes the responses with S6 Appendix providing a full breakdown.

Of the 254 responses, 37 stated they could not quantify the POCT training, with a further 66 reporting no POCT training. In the remaining 151, a range of 0–450 hours was dedicated to POCT training Overall, the median (IQR) hours for teaching POCT was 3 hours (0–7.50) for NHS clinical laboratories. In cases where a range of hours was provided by a respondent, the maximum value was used for consistency.

Fig 3, shows the different training methods used in laboratory teaching of POCT, in response to question 4 of survey 2 (Table 1). For a breakdown between the whole trust responses and the different pathology disciplines, see S7 Appendix.

**Table 4. Types of evidence used by United Kingdom clinical laboratories to meet requirements for health and care professions council standards of proficiency 14.16 and 14.26.**

| Evidence Type | Number of times used | % |
|---|:---:|:---:|
| Reflective statement | 134 | 22.7 |
| Competency assessment | 96 | 16.2 |
| Questions & Answer | 80 | 13.5 |
| Witness statement | 67 | 11.3 |
| Essay/review | 57 | 9.6 |
| Annotated work | 45 | 7.6 |
| Practical | 28 | 4.7 |
| Audit | 26 | 4.4 |
| Case study | 21 | 3.6 |
| Lecture/tutorial | 20 | 3.4 |
| Personal statement | 6 | 1.0 |
| Self-directed learning | 5 | 0.8 |
| University work | 4 | 0.7 |
| Journal club | 1 | 0.2 |
| MDT (multidisciplinary **team meeting**) | 1 | 0.2 |

**Table 5. Summary of hours dedicated to teaching point of care testing in the national health service.**

| Useable results | Range of hours | Responses with POCT teaching | Responses with no POCT teaching | Unknown hours POCT teaching | Median (hours) and interquartile range [Q1-Q3] |
|---|---|---|---|---|---|
| 254 | 0–450 | 151 | 66 | 37 | 3 [0–7.5] |

POCT: point of care testing

This question yielded 225 usable responses. Several responses listed multiple training methods. As seen in Fig 3, practical training and lecture-based training was the most popular training method, with 119 and 130 responses respectively of the 372 overall responses. Thirty-four responses listed no training in POCT with 6 citing learning at university as the method for POCT training.

Question 5 of survey 2 (Table 1) asked if there was any rotation into the POCT section or department for the trainee/student. One hundred and thirty-four laboratories responded to this question with 90 responding for the whole trust. The remaining laboratories gave discipline specific information, resulting in 253 useable responses (see Table 6).

Of the 253 responses, 19 stated they couldn't quantify how much time was spent on POCT rotation. One hundred and fifty-four responses stated that no POCT rotation took place, while a further 26 responded, not applicable. The range of hours on rotation was 0–450 hours. The

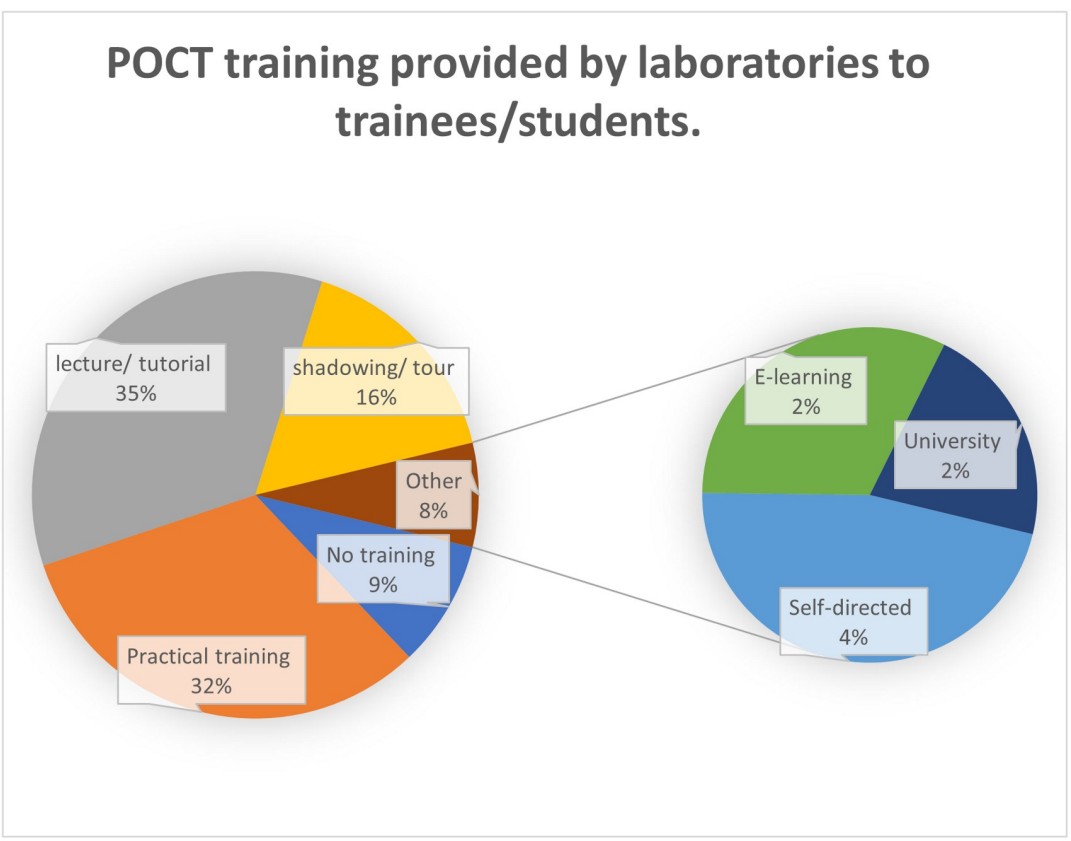

**Fig 3. The different training methods used by United Kingdom national health service laboratories for teaching and training of point of care testing in 2018.**

**Table 6. Summary of time spent on point of care testing rotation in United Kingdom laboratories.**

| Useable results | Range of hours | Responses with POCT rotation | No POCT rotation | N/A | Unknown | Median (hours) and Interquartile range [Q1-Q3] |
|---|---|---|---|---|---|---|
| 253 | 0–450 | 54 (21%) | 154 (61%) | 26 (10%) | 19 (8%) | 0 [0–3.75] |

POCT: point of care testing

median (IQR) time spent in a POCT rotation was 0 (3.75) hours. For a full breakdown of responses, see S8 Appendix.

## 4 Discussion

The purpose of this research was to characterize and analyse what, and how much, POCT is taught at undergraduate level and to those undertaking the IBMS certificate of competence to enter the HCPC register as a BMS.

Both surveys, sent via a FOI request, were sent before the new QAA bench marking statement for biomedical sciences was released in late 2019, so any university seeking accreditation 2020 onwards will have to prove they meet the new QAA statement. This development is bound to affect the amount of POCT taught on these programs.

The first survey is useful to gauge the baseline teaching of POCT at universities. It is worth noting that only 8 of the universities gave LOs specifically describing POCT activities, just 9% of the overall LOs. The remaining 21 universities gave generic laboratory LOs. The remainder of universities provided no LOs to students. From this evidence we can infer that the majority of universities that teach POCT only do so in a non-standardised manner. Our study revealed that only 61% of LOs were measurable. Schoepp, in their review of LO in 10 leading universities in 2017, found that only 54% of LO were measurable [21]. The verbs were assigned to all 6 levels of the blooms taxonomy, with the LO spread throughout the levels, except evaluation. Schoepp, also mapped the LOs to all levels, however, they found that the middle order, application and analysis, dominated [21]. Thirty-one percent of the LOs used action verbs from the avoid list and 66% contained multiple outcomes, therefore not meeting the described structures for a good LO. It is therefore important that universities review their LO so that future employers will have a clear idea what their trainees/students have been taught. The addition of POCT to the requirements of degree accreditation will require universities to include more POCT specific LOs to their curriculum. We suggest that universities engage with clinical laboratories when writing POCT specific LOs to reflect current practice and refer to both the HCPC standards and QAA benchmark statement for guidance.

Twelve of the universities responded that they had no POCT teaching on their courses and a further nine could not give an exact teaching time as POCT was embedded within the curriculum. The median teaching time was 2 hours, and whilst the 2019 QAA benchmark does not stipulate teaching hours, this appears low. The increasing scope of POCT, across the range of disciplines of pathology, merits inclusion across the biomedical science degree curriculum. The LOs provided were mostly generic and were not reflective of the range of POCT equipment available.

From the university's perspective, not all graduates from accredited biomedical science degrees become HCPC registered BMS. Therefore, as they only had to meet the 2015 QAA benchmark statement for IBMS degree accreditation, fitting in all topics relevant to BMS may not have been a priority or been able to fit into their teaching program.

The second survey sent to the NHS laboratories assessed trainees/students undertaking the COC to obtain the core educational requirements as stated by the HCPC.

The training pathway is based on trainees/students undertaking the COC to have the core educational requirements as stated by the HCPC. As the evidence from the first survey sent to the universities showed that a substantial portion of the graduates will not receive much, if any, teaching on POCT then one way to counter this would be to formalize the POCT training of graduates that do become trainee BMS.

The second survey to the NHS clinical laboratories asked the question of how many hours are dedicated to POCT teaching. Forty-one percent of the returns stated that they either had no POCT training or the POCT teaching was unknown. This is a surprising finding, as all trainee BMS require POCT teaching/training to meet the HCPC requirements for registration. This also means that due too no, or often limited, teaching hours of POCT at university, a large percentage of newly qualified BMS will not have the skills or knowledge to meet the requirements for the increased use of POCT. In their national review of English pathology, Lewis et al. [38] looked at the percentage of hospital coagulometer, community glucose and community INR tests were done with laboratory oversight. They found that 24%, 65% and 69% respectively of these tests were outside of laboratory oversight and recommend "*All POCT, wherever it is undertaken, is performed to these agreed quality standards, and under a governance structure that is linked to and supported by an accredited lab*"

A subsequent question asked what types of POCT training was provided in the laboratory for trainees/students. The two most used training techniques were lecture based and practical demonstration, 35% and 32% respectively. It is interesting to note that the two HCPC standards relating to POCT have different outcomes. Standard 14.16 expects BMS to have knowledge of the role and responsibilities of the laboratory with respect to POCT, whilst standard 14.26 expects BMS to have a practical knowledge of POCT analysis. A combination of the two teaching methods therefore seems appropriate.

The final question of the second survey asked about secondments to a POCT section or department. This question relates to the 'hands-on' POCT experience that biomedical science trainees experience. Only 21% of respondents provided the designated hours for a secondment. Therefore, 79% of trainees will have no practical experience of POCT in a clinical environment once qualified. This is backed up by the response to question 2 of survey 2 (Table 1), asking for the types of evidence used to meet the two HCPC standards with POCT in. In 21%, the evidence related to 'hands on' experience, competency assessments and practical. This further reinforces the possibility of a gap in knowledge in POCT for newly qualified BMS.

The question now becomes—does the potential gap in POCT teaching and training matter? This research shows, that in most cases, the amount of POCT taught at both undergraduate level and in pathology laboratories is limited. This research also revealed that in the universities where POCT is taught, it is largely done so in a non-specific or measurable way. With the rise in POCT being supported by pathology, both teaching and training of POCT will have to be formalized, in order to equip personnel to meet this demand.

This work provides the baseline for POCT teaching and training in the UK prior to the change in QAA benchmark statement for biomedical science. The challenge now is for universities providing IBMS accredited degrees to provide this extra teaching. These universities already have close links with pathology laboratories as part of their teaching and consideration should be given to utilizing POCT specialists to support university teaching of POCT.

The research also highlights the lack of specific POCT teaching. The IBMS launched a qualification in 2019 in POCT, aimed at BMS working, or with an aspiration to work in POCT. This was in response to increased calls for a specific qualification [39]. The COVID-19 pandemic has put a spotlight on POCT due to the use of COVID -19 testing kits and will result in increased exposure to POCT in health systems [40]. How this impacts the training and education of laboratory staff in POCT will take some time to discover.

The addition of POCT into the QAA benchmark statement and therefore to the IBMS degree accreditation process will impact universities providing IBMS accredited undergraduate degrees. Universities, especially those identified having no POCT teaching, will have to evaluate their curriculum in light of these changes, if they wish to remain accredited. One possible source of increased teaching support will be local pathology departments. Whilst the data suggests a lack of POCT training in NHS laboratories for newly qualified BMS, there are some clear examples where POCT training is provided using a range of training situations and experience.

Whilst this is the first study to review the curriculum for POCT in Laboratory scientists, earlier research in medical [41] and public health curriculums [42] have also reviewed POCT and also found a lack of POCT teaching despite the increased use of POCT in their fields.

### 4.1 Limitations

One of the limitations of this study is that the FOI request only went to NHS laboratories. As private businesses are exempt from the FOI act, they were excluded from the study. Eight of the respondents stated that their pathology services were run by private laboratories and so did not provide any data. These 8 responses were counted amongst the unusable returns. Another limitation is that the information provided depends on the respondent's interpretation of the request. The FOI acts also allow respondents not to provide the information if it breaches conditions of the acts, such as cost. Two of the respondents used exemptions explicitly and several more stated that they didn't hold the information in response to individual questions.

### 4.2 Conclusion

In summary, it appears that the amount of POCT taught in UK universities is extremely variable and that in the majority of universities it is taught in generic rather than specific terms.

For UK laboratory training in POCT, the trend is to provide just enough training to meet the HCPC standards, with many laboratories having no rotations in to POCT departments for practical work experience in POCT.

Further research is needed to see the impact of both an increased use of POCT and the QAA benchmark changes, in the training and teaching of POCT to students and Trainees in the UK.

## Supporting information

**S1 Appendix. Examples definitions of a learning outcome.**
(DOCX)

**S2 Appendix. Suggested structures of course learning outcomes.**
(DOCX)

**S3 Appendix. Count of verbs used in learning outcomes.**
(DOCX)

**S4 Appendix. Count of action verb by level of blooms taxonomy.**
(DOCX)

**S5 Appendix. Response to question 2 of survey 2 regarding types of evidence to meet health and care professions council standards 14.16 and 14.26.**
(DOCX)

**S6 Appendix. Response to question 3 of survey 2 regarding hours trained in point of care testing.**
(DOCX)

**S7 Appendix. Response to question 4 of survey 2 regarding training methods for point of care testing in clinical laboratories.**
(DOCX)

**S8 Appendix. Response to question 5 of survey 2 regarding rotations into a point of care testing section/department.** POCT: point of care testing.
(DOCX)

## Author Contributions

**Conceptualization:** Lee Peters.

**Supervision:** Ana Sergio Da Silva, Philip Mark Newton.

**Writing – original draft:** Lee Peters.

**Writing – review & editing:** Lee Peters, Ana Sergio Da Silva, Philip Mark Newton.

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
