## [Decision Letter · Decision Letter 0]

6 Jan 2022

PONE-D-21-35377What is the scope of teaching and training of undergraduate students and trainees in Point of Care Testing in UK universities and hospital laboratories?PLOS ONE

Dear Dr. PETERS,

Thank you for submitting your manuscript to PLOS ONE. After careful consideration, we feel that it has merit but does not fully meet PLOS ONE’s publication criteria as it currently stands. Therefore, we invite you to submit a revised version of the manuscript that addresses the points raised during the review process.

One of the reviewers did not recommend publishing this article. I do however feel that there is merit in this very important topic. Please can you ensure that you complete the corrections. Specifically, while this is primarily about the UK experience, there should be some reference to other settings where POCT has been conducted particularly in view of the small numbers included

We look forward to receiving your revised manuscript.

Kind regards,

Elizabeth S. Mayne, M.D.

Academic Editor

PLOS ONE

Journal Requirements:

2. Please ensure that you have specified (1) whether consent was informed, (2) what type you obtained (for instance, written or verbal, and if verbal, how it was documented and witnessed). If your study included minors, state whether you obtained consent from parents or guardians. If the need for consent was waived by the ethics committee and (3) If you are reporting a retrospective study of medical records or archived samples, please ensure that you have discussed whether all data were fully anonymized before you accessed them and/or whether the IRB or ethics committee waived the requirement for informed consent. If patients provided informed written consent to have data from their medical records used in research, please include this information.

3. We note you have included a table to which you do not refer in the text of your manuscript. Please ensure that you refer to Table 4 and 5 in your text; if accepted, production will need this reference to link the reader to the Table.

Additional Editor Comments:

One of the reviewers did not recommend publishing this article. I do however feel that there is merit in this very important topic. Please can you ensure that you complete the corrections. Specifically, while this is primarily about the UK experience, there should be some reference to other settings where POCT has been conducted particularly in view of the small numbers included.

Reviewers' comments:

Reviewer's Responses to Questions

**Comments to the Author**

1. Is the manuscript technically sound, and do the data support the conclusions?

Reviewer #1: Yes

Reviewer #2: Partly

2. Has the statistical analysis been performed appropriately and rigorously? 

Reviewer #1: I Don't Know

Reviewer #2: No

3. Have the authors made all data underlying the findings in their manuscript fully available?

Reviewer #1: Yes

Reviewer #2: Yes

4. Is the manuscript presented in an intelligible fashion and written in standard English?

Reviewer #1: Yes

Reviewer #2: Yes

5. Review Comments to the Author

Reviewer #1: The authors need to clarify if the data (training hours for point of care training) was skewed or normal distributed to determine whether mean (SD) (normal distributed) or median (IQR)(skewed data) should be reported.

Reviewer #2: Overall: This is an interesting paper describing how much POCT teaching is being done in the UK universities and pathology departments.However, samples size may be insufficient to come to conclusions. Also, consider additional discussion on how the findings relate to other similar studies. You may consider reducing the introduction section.

Title

Consider writing UK in full.

Abstract

Line 19: How many NHS hospital pathology departments responded (rate in %) vs. non-responders?

Line 21: Write “12” as Twelve of degree………

Line 23: Is this teaching hours per week, month, year or per course? Authors should clarify.

Line 31: Write QAA in full, then abbreviate thereafter.

Introduction

Line 59-64: This paragraph has no reference.

Line 74: Author should insert the reference number 9 or alternative.

Figure 1: Please label as such and include heading.

Line 103: ……… accreditation post 2019 will have to evidence how they meet the new benchmark statement. Missing phrase

Methods

Authors should clarify why they did not use the similar freedom of information requests to both UK universities and pathology departments.

Results

Authors reported means of the hours of POCT teaching. Reporting standard deviation with means will be helpful.

Line 298: Please provide the statistical reasoning for the use of the maximum value when the range of hours was provided.

Line 3: The ‘below’ should be removed. It implies the figure comes after the paragraph. Also, figure 3 is not labelled adequately. One or two sentences explaining the content of the figure will be helpful.

Discussion

Line 355-363. Results of the surveys are reporting in the discussion section. The appropriate section for these information is in the result section.

How do your findings relate to other studies looking at the same topic? No other studies cited in the discussion? Is this because there are no studies that have done looking at the teaching of POCT in the UK or another country? Please clarify.

6. PLOS authors have the option to publish the peer review history of their article (what does this mean?). If published, this will include your full peer review and any attached files.

Reviewer #1: No

Reviewer #2: No

---

## [Author Response · Author response to Decision Letter 0]

23 Feb 2022

Please see response to reviewers.

---

## [Decision Letter · Decision Letter 1]

15 Mar 2022

PONE-D-21-35377R1What is the scope of teaching and training of undergraduate students and trainees in Point of Care Testing in UK universities and hospital laboratories?PLOS ONE

Dear Dr. PETERS,

Thank you for submitting your manuscript to PLOS ONE. After careful consideration, we feel that it has merit but does not fully meet PLOS ONE’s publication criteria as it currently stands. Therefore, we invite you to submit a revised version of the manuscript that addresses the points raised during the review process.

The majority of the reviewers concerns have been addressed. Please just note the minor typographical and grammatical errors highlighted and correct.

We look forward to receiving your revised manuscript.

Kind regards,

Elizabeth S. Mayne, M.D.

Academic Editor

PLOS ONE

Journal Requirements:

Additional Editor Comments (if provided):

The majority of the reviewers concerns have been addressed. Please just note the minor typographical and grammatical errors highlighted and correct.

Reviewers' comments:

Reviewer's Responses to Questions

**Comments to the Author**

1. If the authors have adequately addressed your comments raised in a previous round of review and you feel that this manuscript is now acceptable for publication, you may indicate that here to bypass the “Comments to the Author” section, enter your conflict of interest statement in the “Confidential to Editor” section, and submit your "Accept" recommendation.

Reviewer #1: All comments have been addressed

Reviewer #2: All comments have been addressed

2. Is the manuscript technically sound, and do the data support the conclusions?

Reviewer #1: Yes

Reviewer #2: Yes

3. Has the statistical analysis been performed appropriately and rigorously? 

Reviewer #1: Yes

Reviewer #2: Yes

4. Have the authors made all data underlying the findings in their manuscript fully available?

Reviewer #1: Yes

Reviewer #2: Yes

5. Is the manuscript presented in an intelligible fashion and written in standard English?

Reviewer #1: Yes

Reviewer #2: Yes

6. Review Comments to the Author

Reviewer #1: A track changes word document with correction of grammatical errors has been provided for the authors to review and agree or adjust the changes.

Minor revisions are listed in the reviewers recommendation document for the authors to address.

Reviewer #2: Abstract

Line 19: write NHS in full and abbreviate thereafter.

Line 29: Leave a space between 9% and (13/137)

Line 31: Add comma after scientist. Should read like this: The findings demonstrate that although this is a commonly required skill for biomedical scientists, there is a clear lack of POCT teaching and training in the UK.

Introduction

Line 45-46: Rewrite glucose and pregnancy in lower cases. Rewrite INR in full and abbreviate thereafter.

Line 57-58: Consider rephrasing this sentence and including that POCT as a method to prevent unnecessary hospital admission. The NHS England long-term plan (7) discusses the use of POCT as a method to prevent unnecessary hospital admissions.

Line 72: Footnotes misplaced

Line 85. Remove extra full stop after Biomedical science (11).

Line 91: Remove extra full stop

Line 122: Change Kennedy, Hyland & Davis to Kennedy et al (19).

Line 134: Add full stop after appendix 2.

Line 158: Change Knight, Stead & Geyton to Knight et al (32).

Material and Methods

Line 182: Change Surveys to surveys.

Line 205: Remove extra space after “searched”.

Results

Line 240: change 97% (173) to 97% (173/179) responded.

Line 252: Add full stop after 2018.

Line 257: Change Los to LOs.

Line 259: Change Forty seven to Forty-seven.

Line 275: change Table 4 Shows to Table 4 shows the evidence used……

Line 302: it is not clear that the sentence “This question yielded 225 usable responses” refers to line 297-299 statement.

Discussion

Line 318: Leave a space between taught and at. Change taughtat to taught at.

Line 326: Remove extra comma after universities.

Line 329: Change to Schoepp et al. in their review of LO in 10 leading……

Line 331: Change to Schoepp et al. mapped the learning…….

Line 364-367: Remove authors. Should read like: In their national of English pathology, Lewis et al. looked at the percentages of hospital coagulometer, community glucose and community INR tests were done with laboratory oversight.

Line 372: Add % after 32.

Line 374: Change eth to the.

Line 423: Lower case Summary

Line 427-430: This paragraph is unnecessary. The aims of the study already articulated under introduction. Please consider removing it.

Reference:

Line 460: Put a space after at https://www.ibms.org/

Line 473: Change it to Jenkins A, Unwind D. How to write learning outcomes. Confirm the website.

7. PLOS authors have the option to publish the peer review history of their article (what does this mean?). If published, this will include your full peer review and any attached files.

Reviewer #1: No

Reviewer #2: No

---

## [Decision Letter · Decision Letter 2]

3 May 2022

What is the scope of teaching and training of undergraduate students and trainees in Point of Care Testing in UK universities and hospital laboratories?

PONE-D-21-35377R2

Dear Dr. PETERS,

We’re pleased to inform you that your manuscript has been judged scientifically suitable for publication and will be formally accepted for publication once it meets all outstanding technical requirements.

Kind regards,

Elizabeth S. Mayne, M.D.

Academic Editor

PLOS ONE

Additional Editor Comments (optional):

Reviewers' comments:

Reviewer's Responses to Questions

**Comments to the Author**

1. If the authors have adequately addressed your comments raised in a previous round of review and you feel that this manuscript is now acceptable for publication, you may indicate that here to bypass the “Comments to the Author” section, enter your conflict of interest statement in the “Confidential to Editor” section, and submit your "Accept" recommendation.

Reviewer #1: All comments have been addressed

Reviewer #2: All comments have been addressed

2. Is the manuscript technically sound, and do the data support the conclusions?

Reviewer #1: Yes

Reviewer #2: Yes

3. Has the statistical analysis been performed appropriately and rigorously? 

Reviewer #1: Yes

Reviewer #2: N/A

4. Have the authors made all data underlying the findings in their manuscript fully available?

Reviewer #1: Yes

Reviewer #2: Yes

5. Is the manuscript presented in an intelligible fashion and written in standard English?

Reviewer #1: Yes

Reviewer #2: Yes

6. Review Comments to the Author

Reviewer #1: All comments and corrections have been addressed.

Minor grammatical changes have been corrected in the uploaded attachment.

Reviewer #2: Abstract

Line 25: Put a full stop between “450 hours and The median time…….”.

Line 30: Change number 8 to eight

Introduction

Line 47: Change the font for severe acute respiratory syndrome coronavirus 2 to similar to the rest of the text.

Line 65: Change Council to lower case

Line 112: Place the full stop after the reference. Should read like this: “in light of recent advances in practice” (11).

Results

Line 254: Write IQR in full and abbreviate thereafter.

Table 3: Write the level in upper case

Supporting documents

Appendix 3: Row labels start with the lower while Appendix 4 start with upper case. I suggest you make row labels similar.

Appendix 4: Change these row labels: ARRANGE, RELATE, SELECT and know to Arrange, Relate, Select, and Know, respectively.

7. PLOS authors have the option to publish the peer review history of their article (what does this mean?). If published, this will include your full peer review and any attached files.

Reviewer #1: No

Reviewer #2: No

---

## [Editor Report · Acceptance letter]

24 Jun 2022

PONE-D-21-35377R2 

What is the scope of teaching and training of undergraduate students and trainees in point of care testing in United Kingdom universities and hospital laboratories? 

Dear Dr. PETERS:

I'm pleased to inform you that your manuscript has been deemed suitable for publication in PLOS ONE. Congratulations! Your manuscript is now with our production department. 

Kind regards, 

on behalf of

Dr. Elizabeth S. Mayne 

Academic Editor

PLOS ONE